# Holistic Wisdom from Abrahamic Faiths' Earliest Encounters with Ancient China: Towards a Constructive Chinese Natural Theology

Jacob Chengwei Feng 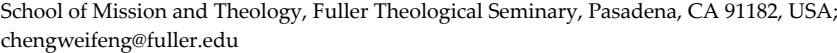

School of Mission and Theology, Fuller Theological Seminary, Pasadena, CA 91182, USA; chengweifeng@fuller.edu

**Abstract:** Philosophies in the East and West have favored wisdom in their search for truths. The Chinese civilization has sought holistic wisdom in its long history of absorbing the Abrahamic faiths since the seventh century. This paper aims to investigate how the Abrahamic faiths have interacted with ancient Chinese culture. In particular, this paper will examine the earliest written records in Chinese of the Luminous Religion (or *Jingjiao*), the earliest Jews in Kaifeng, and the earliest Muslims in China. By analyzing their theology of creation with reference to the Holy Spirit and *qi* (wind/breath/*pneuma*), this paper attempts a constructive Chinese natural theology based on a sympathetic and critical assessment of Alister McGrath's natural theology but makes up for his spirit deficit. This paper argues that the holistic wisdom achieved in the early integration process of the Abrahamic faiths with the Chinese culture is closely intertwined with the Spirit and *qi*, which provides a fruitful ground to construct a Chinese natural theology. The contribution of this paper lies in its original research into the earliest written records of the three Abrahamic faiths in China from the perspective of the doctrine of creation and its relationship with the Spirit and *qi*.

**Keywords:** holistic wisdom; Abrahamic faiths; Jingjiao; Chinese Jews; Chinese Muslims; creation; spirit; *qi*; natural theology

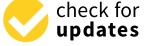



## 1. Introduction

If philosophy means "love of wisdom" in the West, then ancient Chinese philosophers love to "hear the truth" (*wendao* 闻道).[1] Here, Dao refers to truth, the highest wisdom (Zhang 1989, p. 1). Chinese civilization has a high regard for wisdom. Wu Kuang-Ming, a leading philosopher in comparative philosophy and phenomenology, by vigorously engaging in East–West intercultural conversations (Goulding 2013, p. 355), argues that it is necessary to probe "Chinese wisdom alive" today, which "is millennium young, alive today, as a historic alternative to Western culture, ready to engage in inter-enrichment" (Wu 2010, p. xiii). Western culture features alphabetical thinking, digital and abstract, whereas Chinese culture features audio-pictographic thinking. The West has logic–rationality, while China has music–reason, shown by their respective ways of writing out their modes of thinking (Wu 2010, p. xiii). Against Francis Bacon's famous motto, "Knowledge is power", Cao Xing argues that wisdom is the real source of power instead of knowledge. Different peoples have their own unique wisdom and therefore can contribute to the wisdom of humanity from different perspectives (Cao 2005, p. 2). For Cao, "Compared to the wisdom of the West, the wisdom of the Chinese people possesses an inherent peculiarity. The western philosophers are good at analysis, while the Chinese philosophers do well in intuition. Pure analysis is restricted by attention to detail, while intuition concentrates on holism" (Cao 2005, p. 2). Hence, Cao proposes that Chinese philosophy is characterized by its intuitive holism. In making statements as such, Wu and Cao may run the risk of oversimplification; however, their insights are worth noting toward a holistic wisdom that incorporates the wisdom from both the East and the West.

Commenting on the relationship between science and theology, Jürgen Moltmann proposes "a new attempt to see science and theology in the context of the life common to them both [which] is made on the level of *wisdom*" (Moltmann 2003, p. 26, emphasis in original). For Moltmann, "Faith and reason can find each other in the house of wisdom they share, and each of them can contribute its own insights to the building of this house in a culture based on wisdom about life" (Moltmann 2003, pp. 27–28). This paper seeks to explore the "house of wisdom" constructed in the process of the integration of Chinese civilization with the Abrahamic faiths in their earliest encounters in ancient China. I argue that the holistic wisdom achieved in the early integration process of the Abrahamic faiths in China is centered on the Spirit/spirit and *qi* 气 (wind/breath/*pneuma*), which provides a fruitful ground to construct a contemporary Chinese natural theology.

In this paper, I shall first examine the earliest documents found in Chinese that record the life and theology of Islamic, Jewish, and Christian believers. Then, this paper will analyze how these sister faiths interact with the Chinese philosophical concept of Spirit and *qi*. Finally, I will attempt a constructive Chinese natural theology, which is based on Alister McGrath's natural theology but makes up for his spirit deficit. The contribution of this paper lies in its original research into the earliest manuscripts of the three Abrahamic faiths in China from the perspective of the doctrine of creation and its relationship with the Spirit/spirit and *qi*. This paper also attempts to offer some fresh, preliminary proposals toward a Chinese constructive natural theology, hoping to open the door to more meaningful interfaith dialogue and science–theology conversation.

## 2. Earliest Encounters of Abrahamic Faiths in China and Their Textual Records in Chinese

Among the Abrahamic faiths, Christianity entered China the earliest through the East Syriac missionary Aluoben 阿罗本 (or Alopen) along the Silk Road, who arrived in Chang'an, the capital of China, in 635 during the Tang dynasty (618–907).[2] After its initial success in the next two hundred years, *Jingjiao* 景教 suffered a death blow by the Tang Emperor Wuzong 武宗 in 845 in his campaign of anti-Buddhist persecution (Nie 2010, p. 2). The Xi'an Stele, sometimes translated as the "Nestorian" Stele, was unearthed in 1623 or 1625 in the Ming Dynasty. The Stele was erected in 781 and inscribed with texts composed by the Church of the East monk Adam (*Jingjing* 景净) in both Chinese and Syriac, the discovery of which "caused almost as great a stir in the world of learning at that time as did the discovery of the first Dead Sea Scrolls in 1947" (Nicolini-Zani 2022, p. ix). Other documents in the Tang era related to *Jingjiao* written in Chinese are the "Dunhuang" manuscripts discovered from 1908[3] to 1943 by the Japanese Kojima Yasushi[4] and the 815 Luoyang Pillar unearthed in 2006 (Nicolini-Zani 2022, pp. 136–46). In sum, we have in our hands seven (or eight) genuine *Jingjiao* documents in the Chinese language.[5] Also pertinent to this paper is the *Jingjiao* missionaries' scientific learning in astrology, technological skills, and medical science (Feng 2022, pp. 84–88).

Concerning the time and the route of the Jews' arrival at Kaifeng, there has been no scholarly consensus.[6] According to Jordan Paper, the Jewish community in Kaifeng,[7] referred to as the "Chinese Judaism",[8] arrived at Kaifeng around 1000 (give or take a few decades) (Paper 2012, pp. 71–72) and lasted approximately 850 years, while the synagogue lasted for 678 years, from 1163 to 1841 (Paper 2012, p. 3). No explicit writings on theology by the Jews in Kaifeng have come down to us. Compared to the well-documented histories of Buddhism, Christianity, and Islam, there is a paucity of documentation on Judaism. Extremely little exists in Chinese records on the Chinese Jews, and Jewish documentation is limited to the stelae that were on the grounds of the Kaifeng synagogue (Paper 2012, p. 69). Four important stelae have been studied that are labeled with the year of inscription: 1489, 1512, 1663, and 1679.[9] Besides these large stelae with a significant number of texts in the synagogue courtyard, other texts produced by the members of the Kaifeng synagogue community include numerous placards in the worship hall. Fortunately, the Jesuits in the seventeenth and eighteenth centuries were fascinated by the Jews in Kaifeng

and meticulously recorded these inscriptions, almost all of which were lost in the last major flood. Most of the inscriptions were written by the literati of the Kaifeng community, but a few seem to have been written by non-Jewish friends of theirs (Paper 2012, p. 95).

Regarding their content, Andrew Plaks, referring particularly to the 1663 Stela, observes that "even if some or all of these texts were actually composed on commission by non-Jewish literati, according to the established Chinese custom of the time, I still maintain that they reflect primarily Jewish input, whether in the form of preliminary drafts or of discussion with the writers, so that they can be taken as evidence of Jewish thinking on various subjects" (Plaks [1999] 2015, p. 37). This paper shall examine the two earliest Kaifeng stelae: the 1489 Stela[10] and the 1512 Stela,[11] both inscribed in the Ming Dynasty. The text of the 1489 Stela was written by Jin Zhong 金钟, a Chinese Jew who passed the first level of qualification examination into the Official School.[12] The text of the 1512 Stela was penned by Zuo Tang 左唐, a Chinese Jew who was a government official.[13]

The 1489 Stela's original name is *Chongjian Qingzhensi* 重建清真寺, literally "reconstructing the Qingzhen temple". It is well known that *Qingzhensi* is the name of the worshipping house of Chinese Muslims. Against the misunderstanding that the Kaifeng Jews borrowed the name from the Muslims, Wei Qianzhi 魏千志 argues that the Kaifeng Jews named their synagogue *Qingzhensi* and assigned new significance to the phrase *Qingzhen*. *Qingzhen* originally meant "pure and genuine", which was initially unrelated to any religion. The Chinese Jews were the earliest to use *Qingzhen* to describe their own religion. Comparatively speaking, Chinese Muslims began to adopt the name at a later time. Wei quickly adds that this does not mean that the Chinese Muslims borrowed the name of the Chinese Jews. Instead, such naming is the logical result of their own internal development in China, as an adaptation of the earlier names of *Qingjingsi* 清净寺 and *Zhenjiaosi* 真教寺. Therefore, Wei suggests that the Chinese Muslims have also significantly contributed to endowing religious meaning to the phrase *Qingzhen* (Wei 1995).

Several theories have been proposed concerning the exact time when Islam was introduced into China. Wang Lingui 王灵桂 reviews a few scholarly proposals and suggests that no final consensus can be reached due to the paucity of historical records in Chinese and other languages. One trustworthy conclusion is that missionary activities were accurately recorded as late as the beginning of the eighth century.[14] According to the Chinese historical record, in the second year of Yonghui 永徽, Tang Dynasty (651), the Arabian empire sent an envoy to establish contact with China (Gui 2016, p. 10). It took hundreds of years for Islam concepts to be expressed systematically in Chinese as religious classics, which appeared at the end of the Ming Dynasty. The unique characteristics of the Islamic classics in Chinese marked the real birth of Chinese Islam. The borrowing of Chinese terminologies influences the expression of Islamic doctrines, and, more importantly, brings in the interaction between Islam and the Chinese culture represented by Confucianism, Buddhism, and Daoism. This, in turn, has enriched Chinese Islamic doctrines (Yang 2011, p. 1, translation mine). Islam has contributed significantly to Chinese culture. What is noteworthy is the advanced Arabic science and technology in astronomy, medicine, navigation, architecture, and other craftsmanship (Wang 2010, pp. 273–87). The three earliest remaining Islam stelae were inscribed in the Yuan Dynasty (1271–1368) and were found in Quanzhou 泉州 (1350), Guangzhou 广州, and Dingzhou 定州 (Ma 2021, pp. 224–25).

In sum, I have summarized the earliest missionary activities of the three Abrahamic faiths in China and identified their earliest inscription records. This paper now proceeds to analyze some of these records to determine their theology of creation with particular attention to their relationship to the Spirit and *qi*.

### 3. Theologies of Creation of the Earliest Abrahamic Faiths in China and Their Relationship to Spirit and *Qi*

*3.1. Jingjiao's Theology of Creation*

Christoph Baumer compares the different situations of the Western church and the Church of the East missionaries: the former expanded by supplanting theologically weak

religions (such as the Greco-Roman religion) by spreading among illiterate peoples (as in Germania and the British Isles) or by receiving help from civil authorities. At the same time, the latter encountered a highly developed culture and three very vibrant religions or worldviews, which were firmly anchored in the state and among the people. While Confucianism provided the state ideology and marked the machinery of society, Daoism (or Taoism) permeated the self-conception of the people and enjoyed the support of the first Tang emperors. For its part, Buddhism had reached Chang'an about half a millennium before Aluoben and, adopting elements of Taoism, spread rapidly throughout the Middle Kingdom (Baumer 2016, p. 187). Baumer identifies two chief difficulties faced by Aluoben and his successors, namely, explaining the uniqueness of the Savior Jesus Christ and selecting the target audience, since Buddhists, Daoists, and Confucians could not be addressed with the same concepts (Baumer 2016, pp. 187–88). After surveying the seven documents, Baumer characterizes the theology of *Jingjiao* as "a dialogue with Buddhism and Taoism: between orthodoxy and syncretism", which is different from Manichaeism that adapts itself in a chameleon-like manner to its new environment (Baumer 2016, p. 187). Baumer's evaluation does not share the mainstream evaluation that originates from John Foster (Nicolini-Zani 2022, p. 177), who, as early as 1939, argues that

> Terms belonging to the other religions are used throughout [the Tang Christian texts], the Buddhist being the most important. But it is not syncretism. Rather it is a borrowing of terminology, and a relation of doctrine to a familiar background of thought, as the only way of expressing Christian truth in its far-eastern environment.[15]

The theological method of *Jingjiao* is described as "an entirely modern Christian approach to other religions, a method of evangelization based on interreligious dialogue that was virtually unknown in the subsequent history of Christian missionary activity in China" (Nicolini-Zani 2022, p. 178). Their overall theology has been studied and summarized as "orthodox" (Chen 1997, pp. 41–42). To Matteo Nicolini-Zani, their methodology is inreligionization (as opposed to inculturation) due to their "positive view toward another religious vision and practice and occasional adoption of it" (Nicolini-Zani 2022, p. 179). Their assimilation of Buddhist, Daoist, and Confucian terminology is "a selective integration, as it was for Christian literature in the languages of Central Asia" (Nicolini-Zani 2022, p. 185). Concerning their Christology, Roman Malek argues that their "Christian soteriology was formulated within the terminological framework of the Buddhist or Daoist *Weltanschauung* by using the *Dao-* or Buddha/Avalokiteśvara/Guanyin-model to develop a 'Buddho-Daoist' Christology" (Malek 2002, p. 36). Later documents, however, tended to "virtually ignore the crucifixion in favor of a Christology and soteriology that could be more aptly described as Daoist or Manichaean" (Eskildsen 1991, p. 79) or Buddhist (Nicolini-Zani 2022, p. 179).

In evaluating *Jingjiao*'s theology, Nicolini-Zani advises that, first, we "always consider the paucity and one-sidedness of the material available to us. Specific and precise studies on individual documents are therefore to be preferred to general reconstructions" (Nicolini-Zani 2022, p. 181). Second, such an evaluation must be carried out with the dialectic between transmission and transformation. Namely, in every work of cultural *transmission*, which results in the *translation* of original works and the *creation* of new texts, there is always a process of *transformation*, and transformation requires flexibility (Nicolini-Zani 2022, pp. 181–82). Philip Wickeri advises against "assum[ing] Western and European norms of heresy, orthodoxy and syncretism". He insists that we "begin with the East Syrian Christians' understanding of themselves and their mission, and relate this to the context from which they came, as well as the one in which they worked" (Wickeri 2004, p. 52). Nicolini-Zani also notices Chinese culture's "taste for harmony", whose hermeneutical approach makes it possible to overcome the fear of diversity, the suspicion that the other's otherness is a threat to one's own identity. Such an approach resists the Western and Christian hermeneutical categories of syncretism, distortion, and heresy and has been demonstrated in the process of Buddhism's reception to Daoist philosophical experience

and Manichaeism's adoption of the same Buddhist principle of adaptation to the linguistic, cultural, and philosophical context it encountered (Nicolini-Zani 2022, pp. 182–83).

What is of particular interest to this paper and has skipped the attention of most *Jingjiao* scholars is their theology of creation and its relationship with the Holy Spirit, with the exception of Jacob Chengwei Feng's recent publication of *Christianity's Earliest Encounter with the Ancient Techno-Scientific China*. Feng studies the primary texts focusing on the stone inscription of Xi'an due to the text's nature as *Jingjiao*'s statement of faith and argues that their *qi*-tological doctrine of creation contains the following key points. First, this theology is distinctively Christian and Trinitarian in its use of the language "Three-One", "cross", and the Spirit. Second, such a theology integrates the role of the Holy Spirit in creation with the Chinese metaphysical concept of *qi* 气 (or *chi*, breath, pneuma, spirit) by invoking the physical, spiritual, and medical senses of *qi* (Feng 2023, pp. 92–93). Third, the Spirit and wind as a derivative of *qi* are often treated synonymously (Feng 2023, pp. 94–97). Feng proposes that *jing* (as in their name *Jingjiao*) refers to the Holy Spirit in addition to the widely recognized meanings of "light", "grand", and "veneration". Overall, *Jingjiao*'s theology of creation is called *qi*-tological "due to their creative, conceptual imagination by 'dancing' around the Chinese metaphysical concept of *qi*" (Feng 2023, p. 99). Building on top of Feng's proposal, I now examine another reference to creation in *Xuting mishi suojing* 序听迷诗所经 (*Book of the Righteous Meditator*), whose author states that *renren judai Tianzun qi shide cunhuo* 人人居带天尊气始得存活. Namely, "everyone holds within herself the breath of the Honored One of Heaven". One can trace this theological statement to Gen. 2:7. Reflecting on Genesis, Ephrem (d. 373), considered the greatest representative of the Syriac tradition (Brock 2020, p. 3), understands the soul of Adam as having been engendered by God's own breath (Beggiani [1983] 2014, p. 17). Here, one observes that for the *Jingjiao* author, God's *qi*, his own breath, plays a vital role in the creation of humanity.

For Nicolini-Zani, "Physical and *spiritual* balance is given by the proper flow of *qi*. Here it seems to indicate a sort of vital breath … with which God shares life with the first man".[16] This is significant in that first, God's *qi* enlivening humanity reveals the *Jingjiao* author's employment of *qi* at physical, physiological, and spiritual levels while formulating their theology of creation. Second, his *qi*-tological imagination[17] enriches the traditional Chinese concept of *qi* by introducing an additional and vital *spiritual* dimension, significantly contributing to his *qi*-tological theology of creation. At the same time, it is worth noting that the common charge of syncretism in *Jingjiao*'s theology has been generally discredited.[18]

### 3.2. The Theology of Creation of the Chinese Jews

In studying the theology of the Chinese Jews, Paper points out two factors that moved the Chinese Jews moderately away from the theology they brought from Persia: (1) Their mode of thinking came to be far from the Greek-influenced developments emerging at that time in Islamic, Jewish (e.g., Saadia Gaon, Maimonides), and subsequently Christian theology, as Chinese thought is negatively oriented to abstract universals. Therefore, Paper speculates that the Chinese Jews were probably closer to Biblical thought than medieval Christian or Islamic philosophy and theology. (2) Their life experience was very different from that of the Jews in the West, who experienced government-condoned murder, torture, and expulsion, along with limits imposed on their right to occupy and own land. The implicit theology of the Chinese Jews would slowly have come to differ from developments in Europe, the Eastern Mediterranean, and Mesopotamia, while their ritual life and dedication to the Torah and the Talmud remained unchanged (Paper 2012, pp. 17–18). In Paper's estimation, the Jewish theology that developed in China over many centuries emphasized complementary gender equality in relation to a non-anthropomorphic (non-sexed) monotheistic deity, who did not exhibit human traits and was therefore neither punitive nor jealous. Chinese–Jewish theology did not emphasize separation from a hateful world and was similar to the cosmogonic and cosmological understanding of the larger Chinese culture. Thus, Chinese–Jewish theology did not reject this world but fully em-

braced it, understanding humans to be part of nature rather than masters of it. Moreover, its use of Daoist-originated terminology and cosmogony, while not essentially different from medieval Jewish theology, with its non-anthropomorphic understanding of deity, may resonate with those contemporary Western Jews who are oriented toward Daoism and Buddhism (Paper 2012, pp. 23–24).

Of the 1489 Stela, the texts that are related to their theology of creation can be translated as follows:

> The founding Ancestor and Teacher Abraham of Israel was the nineteenth-generation descendant of Pan Gu A-Dan. Since the opening of the heaven and diving of the earth, the Ancestor and Teacher has taught that no images should be molded, no gods and ghosts be worshipped, and no witchcraft be performed. In fact, gods and ghosts are not useful, images bring no blessings, and witchcraft is of no help. Now contemplate this: the heaven is the *qi* that is light and pure, the most Supreme has no match, the *Dao* of heaven lies in *buyan* (wordlessness), practice according to the four seasons and all things will grow. Now observe that things germinate in the spring and grow in the summer, people collect the harvest in the autumn and store them in the winter; those flying and diving, the animals, and the plants flourish and decay, blossom, and fall; those that can be born will be born themselves, those that can transform will transform themselves, those that can take shape will shape themselves, and those that are colorful will take on color themselves.[19]

Based on the text, a few observations can be made regarding its theology of creation. First, the author of the text, Jin Zhong, links Adam in Genesis 1 to the well-known legendary figure Pangu 盘古 in the ancient Chinese creation myth.[20] Second, the frequent references to *zu* 祖 (ancestor), such as *zushi* 祖师 (ancestor and teacher) and *zutong* (the tradition of the ancestor), manifest the Kaifeng Jew's assimilation of the ancient Chinese concept of respecting the ancestors (Ma 2021, p. 229).

Third, the Chinese Jewish understanding of creation is associated with the Daoist concept of *dao* 道, *qi* 气, and *buyan* 不言. Both the 1489 and 1512 Stelae translate the Torah as *Daojing* 道经 and therefore confirm that the Jewish religion conforms to Dao, which can be seen from its respect for *Dao*. Such translation is consistent with the Chinese classic *Daodejing* 道德经, which displays the connotation of the Chinese culture in the Kaifeng Jewish religion. Moreover, Ma Baoquan speculates that the translation of the Torah to *Daojing* is based on the close pronunciation of "To" in Torah and "*Dao*" (Ma 2021, p. 228).

Fourth, the word *zi* 自 appears four times, which indicates the theology's adaptation of the Daoist principle of *ziran* 自然. Lao Zi (or Lao Tzu, 571 BC–471 BC) frames the phrase *ziran* to affirm the autonomy of all things and to safeguard their "self" while insisting on Dao as the Supreme Being in the chain of existence.[21] Paper observes that "[i]n Chinese cosmology, creation is continuous. The movement from nothingness to somethingness to division into two and then the two conjoinings and creating everything else is continually occurring, and in essence, everything continually creates itself. The Chinese term for this is *ziran*, which literally means 'of itself'" (Paper 2012, p. 107). Paper also observes that in quoting the *Analects* (*Lunyu*: xvii, 19), the Kaifeng Jews believe that by the use of *zi* (self), this basic understanding came to Abraham as he mediated on *Tian* (God) and, upon realizing this profound "mystery" (*xuan* 玄), founded Judaism (Paper 2012, p. 107).

In his speculative theology of the Chinese Jews, Paper suggests that based on their familiarity with the two creation accounts of the Torah, the Chinese Jews would have been more likely sympathetic with the first than the second: "Removing the willed aspect of creation, this version accords with the Chinese understanding … that from the singular Dao arises male Sky and female Earth, female Yin and male Yang, which give birth to all of existence, including humans" (Paper 2012, p. 108). Paper helpfully takes into consideration a Jewish counter-tradition that exists in the Israelite popular religion, which has been abundantly demonstrated by the last several decades of archaeology. The female counterpart of God, Asherah, is Earth as well as the wife of El. She is the deity of fertility and nur-

ture. Despite its controversial nature in contemporary Jewish scholarship, Paper argues that such medieval Jewish understanding would not have been antithetical to the ancient Chinese understanding of the splitting of the singular *Dao* into the complementary pair of Sky and Earth. His essential conclusion is that "[t]he differences between the Chinese understanding of creation and the understanding that the ancestors of the Chinese Jews had when they arrived in Kaifeng may not have been strongly pronounced" (Paper 2012, p. 108). If Paper is correct, then the Spirit and *qi*, considered the power of the universe and the vitality of humanity to be united with human bodies to achieve peace and harmony, would be equally crucial for the early Chinese Jews.

*3.3. The Theology of Creation of the Chinese Muslims*

One of the most important marks of the integration between Chinese Islam and Chinese traditional culture is the formation of the Kalām system, which is characterized by interpreting the Islam doctrines with Confucian thoughts. Its formation took a long time and was gradually established at the end of the Ming Dynasty and the beginning of the Qing Dynasty, represented by the initiation of mosque education and the interpretation of the Qur'an with Confucianism (Wang 2010, pp. 206–7). In order to explore their theology of creation, we will now analyze the text inscribed in the 1348 Stela, which reads:

> Our ancestors have treated all nations as their dwelling places, and never stopped doing good. Their foundation of life is to serve Heaven without setting forth any images.… The Creator cannot be sought after by form and traces. It is a blaspheme if one produces an image, thereby likening the Creator to an object. Only contemplate without an image to express your sincerity. The Creator can be known by the beauty of the remaining customs of the people.[22]

The Chinese Muslims' early account of the theology of creation demonstrates the following features. First, their theology explicitly acknowledges the Creator of the universe, who is invisible and should not be worshipped with any image. Second, the concept of *tian* 天 (Heaven) is taken from the Chinese worldview and worshipped as the Creator. Therefore, it is no wonder that the Qur'an is translated as *Tianzhijing* 天之经 or *Tianjing* 天经. Later stelae have frequent references to the Heaven: "the saints preach the Dao on behalf of the Heaven" (the 1495 Stela), "worship facing the west, invoke the Heaven and celebrate the saints" (the 1453 Stela), "the Upper Heaven is the most respected and greatest, without any comparison or match" (the 1496 Stela), and "worship the Heaven as the Lord" (the 1519 Stela).

Liu Zhi 刘智 (1669–1764) is a typical representative of the Muslim scholars of the Ming and Qing dynasties who unabashedly integrates the basic core and essential principles of the Muslim faith and the metaphysical resources of Confucianism. In his *Tianfang xingli* 天方性理 (or *Caaba and Neo-Confucianism*), Liu proposes that before the creation of the physical world, there existed a *Xiantian shijie* 先天世界 (an earlier world), whose origin is *Wucheng* 无称. He acknowledges that *Taiji* 太极 in *Yijing* 易经 (or Yi Ching), *Wuji* 无极 in Zhou Dunyi 周敦颐 (1017–1073)'s thought, and *Dao* can be considered as the origin of this physical world. Liu creatively developed and expanded the Muslim theology of the emergence of the universe with *qi* as a constitutive element (Ma 2006, pp. 131–32). Likewise, the al-ruh (Holy Spirit) plays an active role with humankind in creation and in revelation. The al-ruh has evidently inspired all the prophets and even Muslim believers according to the Qur'an (Shih-Ching 2006).

In sum, I have analyzed some of the earliest written records of *Jingjiao*, the Chinese Jews, and the Chinese Muslims that contain traces of the theology of creation. I have also identified that in the process of intermingling with the Chinese civilization, each religion not only borrows Chinese terminologies but also integrates in unison the Chinese metaphysical concept of the Spirit and *qi*. Based on such an observation, this paper draws some general contours toward constructing a Chinese natural theology.

### 4. Towards a Constructive Chinese Natural Theology

At the outset, it is essential to point out that this section does not aim at constructing a full-fledged Chinese natural theology. Instead, based on the above analysis of the Abrahamic faiths' doctrines of creation focusing on their engagement with Spirit/spirit and *Qi*, I intend to draw some general and preliminary contours toward such a constructive task.

First, with regard to constructive Christian theology, all theologies are contextual. To use the words of Veli-Matti Kärkkäinen:

> For constructive Christian theology to speak to the issues, questions, and challenges of the pluralistic world, it has to open up to a dialogue with diverse voices from both inside and outside.… [T]he hegemony of aging white European and North-American men—to which company I myself belong!—must be balanced and corrected by contributions from female theologians of various agendas such as feminist, womanist, and *mujerista*; women from Africa, Asia, and Latin America; other liberationists, including black theologians of the USA and sociopolitical theologians from South America, South Africa, and Asia; and postcolonialists, as well as others. (Kärkkäinen 2013, pp. xi–xii)

To the list identified above by Kärkkäinen I shall add "Chinese theology", which is defined by Paulos Huang as "Christian theological reflection on, from, for, and about China broadly defined, as well as its people and culture" (Huang 2022, pp. 3–4).[23] Following Kärkkäinen's proposal, this article opens up a dialogue with a voice from the Global North—the natural theology of Alister McGrath—in order to address the Global South in general and the Chinese civilization in particular. Moreover, it also converses with "voices from outside" by comparing notes with the doctrines of creation from the Chinese Jews and the Chinese Muslims.

Second, in recent decades, there has been a resurgence of interest in natural theology among Catholics and Protestants. Among the Protestant natural theologians, McGrath is considered one of the most prominent contemporary defenders of natural theology (Kojonen 2020, p. 41).

McGrath establishes a theological framework, namely, a Trinitarian natural theology based on the assumption that the universe is "fine-tuned", which mainly refers to the universe's fecundity (McGrath 2009, pp. xi–xiii). One critical claim made by McGrath to provide a "renewed" natural theology is the recognition that the predominant view of the Enlightenment, namely, that "nature" designates an objective reality, is false. Instead, "nature" is an already interpreted entity, the construals of which are essentially tractable and indeterminate, highly susceptible to conceptual manipulation by the human mind (McGrath 2009, p. 6). Therefore, nature "requires *reinterpretation*" by being "seen" in a new way (McGrath 2009, p. 6). Here, the ancient Chinese Jewish theology's adoption of the Daoist concept of *ziran* joins forces with McGrath in refuting Enlightenment thought. Far from an objective reality, the Chinese Jews realize that humanity is not distanced from nature but is an organic part of it due to their ability to give birth: *sheng* 生. The Enlightenment's concept of humanity dominating nature would have been foreign to them.

After an extensive study on a spectrum of approaches to nature, ranging from cosmology through chemistry to evolutionary biology, McGrath discerns that modern science takes a stratified approach to "nature" and argues that there is no reason why an engagement with the quest for beauty in human culture, or the human longing for something unattainable, should not also be seen as integral aspects of natural theology. For him, natural theology is not limited to rational observation and reflection upon the biological or astronomical realms. McGrath adopts Augustine's doctrine of creation, acknowledging its nascent form, to embrace both primordial actuality and emergent potentiality, which provides more room for the future development of natural theology as legitimate and necessary expansion, rather than distortion or subversion, of Christian notions of creation (McGrath 2009, p. 216). McGrath's creative adaptation of Augustine's doctrine of creation is consonant with the Jewish absorption of the Chinese metaphysical concept of *Zi*, which entails the autonomous power of germination within creation.



McGrath's natural theology has enormous theological values in that his approach does not only uphold Christian tradition, and in this case, the doctrine of creation, but also nurtures a Christian way of "seeing" the natural world, a seeing from within the Christian tradition, deriving both its foundations and coherence from a Trinitarian ontology. Such a way of seeing things resonates strongly with our observation and experience of the world. He recognizes that such a natural theology not only possesses an evangelical capacity to explain nature but leads to something much more significant, the capacity to endow life with meaning (McGrath 2009, p. 220). This is significant for Chinese natural theology to stay relevant to the contemporary world while faithfully continuing our mission of transforming lives.

McGrath's suggestion that we reconsider natural theology as a mode of "seeing" nature rather than as an apologetic seems fruitful. It avoids the mistakes natural theologies have made in the past, such as focusing on "gaps" in our understanding or limiting God to a distant watchmaker. Contemporary Chinese theology has largely remained apologetic in its engagement with modern sciences in general and evolution theory in particular.[24]

However, the above comparison of the three Abrahamic faiths' doctrines of creation reveals the "blind spots" of McGrath's natural theology. First, McGrath endeavors to propose a Trinitarian natural theology, but the reference to the Holy Spirit, whether explicit or implicit, is scarce. These will raise suspicions of the "true" Trinitarian nature of his theology. In this respect, *Jingjiao*'s Trinitarian proto-natural theology, with an emphasis on the Spirit and her creative "dance" with *qi* (Feng 2023, p. 92),[25] serves as a solid foundation for constructing a Chinese natural theology. A further investigation of the word *qi* reveals that the word appears ten times in the entire Tang *Jingjiao* corpus,[26] indicating the rich potential of connecting the Holy Spirit and *qi* in the proposed natural theology. One example suffices here. In *Yishenlun* 一神论 (*Discourse on the One God*), *Mishihe* 弥施诃 (Messiah) is said to rise "from death, thanks to the strength of *qi* of the Honored One of Heaven". In the Peshitta English New Testament, Jesus the Messiah is said to be one "who was known by power and by the Holy Spirit as God's Son, and who arose from the dead" (Rom. 1:4) (Kiraz 2020, p. 419). Here, one sees that the *Jingjiao* author divides the phrase *Tianzun qili* 天尊气力 into three parts, *Tianzun*, *qi*, and *li*, and correlates them with God, the Holy Spirit, and power, respectively, thereby equating the Holy Spirit to the breath of the Honored One of Heaven.

To make up the spirit deficit in McGrath's framework, it is worth noting that Jürgen Moltmann likens the *qi* (or *Ch'I*, *Chi*) in *Daodejing* to God's *ruach* in that the latter "is onomatopoeic, echoing the tempest, like Ch'i, but it means both the breath of the eternal God and the vitality of created beings" (Moltmann 2003, p. 191). Likewise, Grace Ji-Sun Kim suggests an understanding of the Spirit as *Chi* to bring believers to a more holistic understanding of pneumatology and combat what is considered to be a limited understanding of the Spirit. Central to her thesis is the concept of Spirit—*Chi*. However, she does not limit comparisons of the Spirit to the Asian understanding of *Chi*, as is found in Taoism, Hinduism, and Buddhism (Kim 2015, p. 136). She also finds comparisons in the life energy *num* of the Kiung San African people, the *nafas* of Islam, the *prana* of India, the *waniya* of the Sioux Native American tribe, the Japanese *Ki*, and the Hawaiian *ha*. For Kim, "many cultures have words to articulate similar ideas of breath, life, and vital energy expressed by the Christian understanding of the Holy Spirit" (Kim 2015, pp. 132–34). Kim's insights provide a pneumatological framework for interfaith and intercultural dialogue, which is affirmed in all three Abrahamic faiths on Chinese soil.

Second, McGrath's natural theology is short of interfaith engagement, which is arguably an indispensable task required by the increasingly pluralistic world. The similarities and differences in the doctrines of creation of the three Abrahamic faiths provide rich "ingredients" for the constructive natural theology. A few examples suffice here: (1) All three faiths' common integration of *qi* adds a hitherto-unknown dimension to the traditional Chinese understanding of *qi*. Conversely, the ubiquitous nature of *qi* in the Chinese civilization poses a challenge for religions to creatively "dance" with it in their incultur-

ation process. (2) Such commonality does not come at the cost of negating the different religious identities. Likewise, the role and function of the Holy Spirit in each Abrahamic faith cannot be confused in terms of their theological significance.

To sum up, based on a critical and sympathetic evaluation of McGrath's natural theology, a Chinese natural theology can be constructed by integrating *Jingjiao*'s proto-Trinitarian theology with a particular emphasis on the Spirit and *qi* as a common denominator among the three Abrahamic faiths in China.

## 5. Conclusions

The Chinese civilization has been striving for holistic wisdom in its long history of absorbing various ingredients from the Abrahamic faiths since the seventh century. In turn, *Jingjiao*, the Kaifeng Jews, and the earliest Chinese Muslims have been shown to effectively absorb the Chinese metaphysical views of *qi* in constructing their theology of creation. I have examined the earliest inscriptions in Chinese of the three sister faiths. By analyzing their theology of creation with reference to the Holy Spirit and *qi* (wind/breath), this paper has attempted a constructive Chinese natural theology, which is based on a sympathetic and critical assessment of Alister McGrath's natural theology and makes up for his spirit deficit. This paper has argued that the holistic wisdom achieved in the early integration process of the Abrahamic faiths with the Chinese worldview is centered on the spirit and *qi*, which provides a fruitful ground to construct a contemporary Chinese natural theology. With the proposed general and preliminary guidelines toward a constructive Chinese natural theology, it is the hope of this author that a door can be opened into a fruitful interfaith dialogue and science–theology conversation.

**Funding:** This research received no external funding.

**Informed Consent Statement:** Not applicable.

**Data Availability Statement:** Data sharing not applicable.

**Conflicts of Interest:** The author declares no conflict of interest.

## Notes

[1]　　The Chinese words/phrases used in this paper are written in the format of pinyin followed by the simplified Chinese characters and, if necessary, characters in Wade–Giles romanization.

[2]　　This group is usually dismissed as "Nestorian" and therefore deemed heretical. However, Brock has strongly argued that the so-called Nestorian church has, in antiquity, preferred to self-describe itself as the "Church of the East". However, the association between the Church of the East and Nestorius is "of a very tenuous nature" and is "totally misleading and incorrect". See (Brock 1996, p. 35; Bantu 2020, p. 202). Lin Ying speculates that besides the Church of the East, another branch of Christianity also from Syria also sent their missionaries, the Fulin monks, or the Melkites, to China in the Tang dynasty. See (Lin 2006). There has been voluminous scholarly works in French, Japanese, English, and Chinese. For a selected bibliography, see (Morris and Chen 2020).

[3]　　Discovered by the French sinologist Paul Pelliot and later catalogued as Pelliot chinois 3847 and abbreviated as P. 3847. P. 3847 contains two texts: the *Hymn in Praise of the Salvation Achieved through the Three Majesties of the Luminous Teaching* (*Jingjiao sanwei mengdu zan*) and the *Book of the Honored* (*Zunjing*), plus some final notes.

[4]　　The Kojima manuscripts include Kojima manuscript A: *Hymn of Praise to the Most Holy One of the Luminous Teaching of Da Qin, through Which One Penetrates the Truth and Turns to the Doctrine* (*Da Qin jingjiao dasheng tongzhen guifa zan* and Kojima manuscript B: *Book of the Luminous Teaching of Da Qin on Revealing the Origin and Reaching the Foundation* (*Da Qin jingjiao xuanyuan zhiben jing)*. See (Nicolini-Zani 2022, p. 147). The authenticity of the Kojima manuscripts has been challenged. See (Nicolini-Zani 2022, pp. 154–55).

[5]　　Nicolini-Zani specifies seven documents, including the Xi'an Stele and six scrolls including from Cave 17 in Dunhuang, sealed in 1036, as presumably the other two do as well (two additional fragments are hotly disputed). TEXT A: "*Stele of the Diffusion of the Luminous Teaching of Da Qin in China*" (*Da Qin Jingjiao liuxing Zhongguo bei* 大秦景教流行中國碑); TEXT B: (1) "*Hymn in Praise of the Salvation Achieved through the Three Majesties of the Luminous Teaching*" (*Jingjiao sanwei mengdu zan* 景教三威蒙度讚); (2) "*Book of the Honored*" (*Zunjing* 尊經); TEXT C: "*Discourse on the One God*" (*Yishen lun* 一神論): I. "*Discourse on the One Godhead*" (*Yitian lun diyi* 一天論第一) II. "*Metaphorical Teaching*" (*Yu di'er* 喻第二) III. "*Discourse of the Honored One of the Universe on Alms-giving*" (*Shizun bushi lun disan* 世尊布施論第三); TEXT D: "*Book of the Lord Messiah*" (*Xuting mishisuo jing* 序聽迷詩所經); TEXT E:

"*Book on Profound and Mysterious Blessedness*" (*Zhixuan anle jing* 志玄安樂經); TEXT F: "*Book of the Luminous Teaching of Da Qin on Unveiling the Origin and Attaining the Foundation*" (*Da Qin Jingjiao xuanyuan zhiben jing* 大秦景教宣元至本經). See (Nicolini-Zani 2022, pp. 193–303). There are other methods of classification. For example, Nie Zhijun simply collects eight texts in Chinese. The eight texts are (1) TEXT D; (2) TEXT C; (3) TEXT E; (4) TEXT B (1); (5) TEXT B (2); (6) TEXT F; and (7) Sutra pillar of the *Book of the Luminous Teaching of Da Qin on Unveiling the Origin and Attaining the Foundation* (*Da Qin jingjiao xuanyuan zhiben jingchuang ji* 大秦景教宣元至本經幢記); (8) TEXT A. See (Nie 2010, pp. 330–66).

[6] Li Dawei summarizes five views concerning the time of the Jews' arrival at Kaifeng: (1) pre-Zhou Dynasty (1100 BC–256 BC), which is highly unlikely; (2) Zhou Dynasty, also highly unlikely; (3) between the Jews' captivity in Babylon and mid-second century B.C.; (4) Han Dynasty (202 BC–220 AD); (5) Tang Dynasty (618–907); (6) Song Dynasty (960–1279). Various scholars have studied when the Jews left their native land and when they entered into China based on their correlation between the Jews' experience in their native land and the inscriptions, legends, and oral histories of the Kaifeng Jews. The majority of the scholars suggest that the Jews arrived at Kaifeng once for all, while other scholars, such as Pan Guangdan, insist that the Jews arrived at different times. See (Li 2015).

[7] For the best succinct history of the Chinese Jews, see (Leslie 1972). For a work that focuses on the European reaction to the "discovery" of the Chinese Jews, see (Pollak 1983). For the most recent, and by far the most readable account, see (Xu 2003). For a recent work in Chinese, see (Song 2012).

[8] According to Jordan Paper, today many assume that the term "Chinese Judaism" refers to the century and a half of European Jews living in China—that is, "Judaism in China"—rather than the "Chinese Judaism" of the last millennium. See (Paper 2012, p. 4).

[9] Two stelae were from the Ming Dynasty: (1) the 1489 Stela, *Chongjian qingzhensi ji* 重建清真寺记, also called *Hongzhi bei* 弘治碑; and (2) the 1512 Stela, *Zunchong daojingsi ji* 尊崇道经寺记, also called *Zhengde bei* 正德碑. The other two were from the Qing Dynasty: (3) the 1663 Stela, *Chongjian qingzhensi ji* 重建清真寺记, also called *Kangxi bei* 康熙碑; and (4) the 1679 Stela, *Citang shugubei ji* 祠堂述古碑记.

[10] The 1489 Stela is currently preserved in the Kaifeng Museum. The text was compiled by Chen Yuan. See (Chen 1982, pp. 256–59).

[11] The 1512 Stela is currently preserved in the Kaifeng Museum. The text was compiled by Chen Yuan. See (Chen 1982, pp. 260–62). Chen's text was verified by Ma Baoquan. See (Ma 2021, p. 224).

[12] Jin Zhong's academic title is *Kaifengfu Ruxue zengguang shengyuan* 开封府儒学增广生员.

[13] Zuo Tang's official title is *Sichuan buzhengsi youcanyi* 四川布政司右参议, an advisor of the governor of Sichuan province.

[14] The proposed dates are (1) Kaihuang era (581–600) of Sui Dynasty; (2) Wude era (618–626) of Tang Dynasty; (3) early Zhenguan era (627–649); (4) the second year of Yonghui era (651); and (5) the beginning of the eighth century. See (Wang 2010, pp. 106–11).

[15] See (Foster 1939, p. 112). Tang Li argues similarly: "Even though Nestorians adopted many Buddhist and Daoist phrases in their texts, syncretism should not be considered a serious case". See (Tang 2002, p. 142).

[16] For Nicolini-Zani, physical and spiritual balance is given by the proper flow of *qi*. Here, it seems to indicate a sort of vital breath (that of Genesis 2:7?) with which God shares life with the first man. See (Nicolini-Zani 2022, p. 266), emphasis added.

[17] Amos Yong speaks of pneumatological imagination as the logic of Pentecostal theology. See (Yong 2020).

[18] Johan Ferreira has expressed it in a clear and definitive manner: "Contrary to common opinion, the theology of the Tang Chinese church was not an aberrant form of Christianity with only internal or syncretistic concerns, it was consistent with traditional orthodoxy" (Ferreira 2014, p. 337). Such a statement is also accepted with absolute certainty by Matteo Nicolini-Zani (Nicolini-Zani 2022, p. 119).

[19] My translation of the first several lines of the 1489 Stela is based on the text compiled by Chen Yuan. See (Chen and Chen 1981, pp. 65–68).

[20] For a discussion on Pangu and the origin of universe, see (Wu 2011).

[21] Wang argues Lao Zi introduces a new worldview in his framing of the concept of *zi ran* 自然. See (Wang 2019).

[22] The Chinese text is recorded in (Yu and Lei 2001, chp. 4), translation mine.

[23] Theologians such as Wing-hung Lam, Liu Xiaofeng, He Guanghu, Chloë Starr, and Alexander Chow have contributed to such a constructive task. See (Lam 1983; Liu 2000; He 1996; Starr 2016; Chow 2018).

[24] For a theological assessment of how the theory of evolution impacted Chinese theology, see (Feng 2022, pp. 301–5).

[25] Here, I follow the East Syrian—and hence, *Jingjiao*—tradition, which spoke of the Holy Spirit as a feminine figure. Johannes van Oort argues that the earliest Christians—all of whom were Jews—did the same. Such an ancient tradition was kept alive in East and West Syria, up to and including the fourth century Makarios and/or Symeon, who even influenced "modern" Protestants such as John Wesley and the Moravian leader Count von Zinzendorf. It is concluded that, in the image of the Holy Spirit as woman and mother, one may obtain a better appreciation of the fullness of the Divine. See (Van Oort 2016).

[26] Once in "*Stele of the Diffusion of the Luminous Teaching of Da Qin in China*" (*Da Qin Jingjiao liuxing Zhongguo bei*), four times in "*Discourse on the One God*" (*Yishen lun*), three times in "*Book of the Lord Messiah*" (*Xuting mishisuo jing*), and twice in "*Book of the Luminous Teaching of Da Qin on Unveiling the Origin and Attaining the Foundation*" (*Da Qin Jingjiao xuanyuan zhiben jing*).

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
