# Peer review of "Holistic Wisdom from Abrahamic Faiths’ Earliest Encounters with Ancient China: Towards a Constructive Chinese Natural Theology"

_religions, doi:10.3390/rel14091117_

Round 1

Reviewer 1 Report

The paper's emphasis on the earliest written records of the three Abrahamic faiths in China, specifically in the context of the doctrine of creation and its connection to the Spirit and qi, is a contribution to the existing body of knowledge. By examining historical sources and presenting a synthesis of ideas, the paper adds depth to the understanding of cross-cultural interactions and theological developments. The author's insights into the intercultural and interreligious dynamics between Abrahamic faiths and ancient China align well with the themes of the Special Issue.

The article generally maintains a clear and coherent structure, effectively guiding readers through the analysis. The integration of various faith traditions, philosophical concepts, and theological ideas is handled skillfully. However, certain sections might benefit from further clarification or elaboration, particularly when connecting the Holy Spirit and qi in the proposed natural theology framework.

Second, both Western and Eastern ideas are a very complex system with many diverse characteristics, so the article should be more precise and careful in making conclusions about them, avoid statements that lack specifics and are too absolute. Such as “The West has logic-rationality, while China has music-reason, shown by their respective ways of writing out their modes of thinking (Wu 2010, xiii).” Although many points of view come from the author's citation, Such as For Cao, “Compared to the wisdom of the West, the wisdom of the Chinese people possesses an inherent peculiarity. The western philosophers are good at analysis, while the Chinese philosophers do well in intuition. Pure analysis is restricted by attention to detail, while intuition concentrates on holism” (Cao 2005, 372). But for cross-cultural readers, who is Cao Xin? Why is his point of view correct and representative? Since the author has chosen to use a large number of other people's conclusions to support his own argument, and these circumstances need to be further explained.

This paper seems to lack a detailed literature review, so what the authors think “The contribution of this paper lies in its original research into the earliest manuscripts of the three Abrahamic faiths in China from the perspective of the doctrine of creation and its relationship with the Spirit/spirit and qi” is not so convincing. Jingjiao, Abrahamic Faiths , or the encounter between Christianity and China more generally, is not an obscure academic field, and many scholars have done relevant research, such as Professor Wu, Chang Shing from Taiwan, who has published a monograph of nearly one million words. Unless you can definitively prove that these historical documents were first excavated and cited by you. Otherwise, you need to do a detailed literature review to highlight the differences and contributions of your research and previous studies.

Finally, the language and presentation of the article are generally appropriate for an academic journal. The writing style is scholarly and demonstrates a strong command of relevant terminology. Nevertheless, some sentences could be refined for greater clarity, and proofreading for grammatical and typographical errors would enhance the overall readability. Especially as an English study based on a large number of Chinese materials, it is recommended to seek the help of professional editors to discuss and consider the translated parts.

Reviewer 2 Report

See enclosed file 

Round 2

Reviewer 1 Report

Accept in present form

Author Response

Dear Reviewer,

Thank you for your extremely useful feedback.

Blessings!

Reviewer 2 Report

see attached file

Author Response

Dear Reviewer,

Thank you for second-round feedback, which is highly appreciated. Please see attached for my revision and response.

Round 3

Reviewer 2 Report

I read the new version of the paper and the author's response to my comments. I want to thank the author for the courtesy to take my comments seriously. Despite the changes, I am convinced that the author is trying to keep together two distinct articles.

One is the inculturation of the three Abrahamitic faiths in China with specific reference to natural theology. This article, to be completed, require a further section that compare the nature and evolution of the three different natural theologies in China.

Another article is the dialogue between the Chinese natural theologies and McGrath’s natural theology. I think this is where the author wants to be, but more work is necessary to establish this dialogue. Originally, the author's idea was to summarize the three (Christian, Muslim, and Jewish) natural theologies into one, and use this one to correct McGrath. But this is syncretism. At this point each natural theology deserves a distinct dialogue with McGrath. Maybe the author can select one author for each Chinese theology and place him/her in dialogue with McGrath. 

The real problem is the idea to compare a collective theology (a theology collectively developed in China) and the theology of a single author. I do not know, I am not sure it works. The whole idea to use Chinese, or German, or Canadian natural theology to correct McGrath seems to me strange. If the author wants to proceed on this path, he/she cannot simple offers his/her voice, but also the voice of the Chinese theologians. 
